# Effect of Phosphorus Content on Magnetic and Mechanical Properties of Non-Oriented Electrical Steel

**DOI:** 10.3390/ma15186332

**Published:** 2022-09-13

**Authors:** Qinyu He, Yulong Liu, Chengyi Zhu, Xiaohui Xie, Rong Zhu, Guangqiang Li

**Affiliations:** 1The State Key Laboratory of Refractories and Metallurgy, Wuhan University of Science and Technology, Wuhan 430081, China; 2Key Laboratory for Ferrous Metallurgy and Resources Utilization of Ministry of Education, Wuhan University of Science and Technology, Wuhan 430081, China

**Keywords:** non-oriented electrical steel, phosphorus, magnetic property, mechanical property

## Abstract

The effect of target phosphorus (P) content on the precipitates, microstructure, texture, magnetic properties, and mechanical properties of low-carbon (C) and low-silicon (Si) non-oriented electrical steel (NOES) was investigated and the influence mechanism was clarified. The results indicate that the precipitates in the steels are mainly aluminum (Al)-manganese (Mn)-Si-bearing complex nitrides ((Al,Si,Mn)*_x_*N*_y_*) and P-bearing complex nitrides ((Al,Si,Mn)*_x_*N*_y_*-P). Increasing target phosphorus content in the steels decreases (Al,Si,Mn)*_x_*N*_y_*, and increases (Al,Si,Mn)*_x_*N*_y_*-P. The number density of the precipitates is the lowest, and the average size of the precipitates and grain size of the finished steel is the largest in the samples with target P content at the 0.14% level (0.14%P-targeted). The average grain size and microstructure homogeneity of the steels are influenced by the addition of phosphorus. The content of the {111}<112> component decreases, and the favorable texture increases after phosphorus is added to the steel. The magnetic induction of the steel is improved. Grain refinement and microstructure inhomogeneity lead to an iron loss increase after target phosphorus content increases in the steel. The best magnetic induction *B*_50_ is 1.765 T in the 0.14%P-targeted samples. The tensile strength and yield strength are improved owing to solid solution strengthening and the grain refinement effect of phosphorus added to the steels.

## 1. Introduction

Non-oriented low-carbon (C) low-silicon (Si) electrical steel with Si or Si and aluminum (Al) content less than 1.0% is widely used as the iron core material for all kinds of small-power motors, magnetic ballasts, and small power transformers [1]. Low-grade non-oriented electrical steel (NOES) has excellent cost performance because of its high magnetic induction, relatively simple manufacturing process, and low manufacturing cost. With the increasing demand for low carbon, energy savings, and environmental protection, further improving the present performances of various electrical equipment is required by the end-users. For this reason, manufacturers of electrical equipment such as motors and transformers require NOES to have excellent magnetic properties, high magnetic induction, low iron loss, and low magnetostriction, which will meet the requirements, such as miniaturization, lightweight, high efficiency, low loss, and low noise, of electrical equipment. On the other hand, NOES will withstand shearing and stamping before laminating when the iron core of the motor is manufactured, and the iron core needs to resist the centrifugal force caused by high-speed rotation when the motor is in operation. Therefore, the mechanical properties of NOES should be improved to meet the end-users’ requirements for motor quality and extend the service life of motors, as well as ensure the magnetic properties.

The composition of the alloying elements and impurities (nanoscale precipitates and inclusions in the steel) of the steel are the main metallurgical parameters that impact the magnetic and mechanical properties of electrical steel [2,3,4,5,6,7]. Phosphorus has a larger atomic radius than iron and silicon, and phosphorus has a good solid solution strengthening effect, which can significantly increase the hardness and improve the performance of steel. The mechanical properties of NOES can be enhanced by the use of the solid solution strengthening effect of alloying elements such as phosphorus, but if the phosphorus content in the steel is too high, particularly for the steel with very low C content, the cold workability of the steel will deteriorate and lead to the finished product’s brittle failure. Phosphorus is a ferrite-forming element. Phosphorus added into NOES can increase the resistivity, decrease the austenite phase area, encourage grain development, and minimize eddy current loss to a certain extent. The magnetic induction is increased by benefitting from P segregation along the grain boundaries by raising the {100} component and decreasing the {111} component in the steel [1]. Ray et al. [8] added 0.17%P to NOES, and they found that the carbide precipitation and growth were effectively hindered, the iron loss was reduced, and the magnetic aging was mitigated. Lee et al. [9] added 0.2%P to the steel containing 3% Si, and the cubic texture {100}<001> increased, while the {111}<112> texture decreased in the steel, which improved the magnetic induction of the products. The grain size of the finished steel becomes smaller, because of the segregation of phosphorus at the grain boundaries, which lowers the iron loss of the steel. The yield strength of the steel is raised and the mechanical properties of the finished steels are enhanced by the grain refining and solid solution strengthening effects of phosphorus. However, it was also reported that the phosphorus content in the more Si-Al-bearing steels should not exceed 0.02%, as it is prone to segregation at the grain boundaries, inhibits grain development, and results in brittle cracking in steel plates [10]. Tanaka et al. [11] reported that a favorable texture and improved magnetic induction of the finished steel were obtained, owing to the segregation of phosphorus at the grain boundaries. The magnetic induction of the P-added steel decreased only a little after the cold rolling reduction rate increased, while that decreased dramatically for the steel without P added. Lee et al. [12] found that the recrystallization of the steel was retarded and the grain coarsening rate increased owing to the segregation of phosphorus at the grain boundaries. A larger grain size and favorable texture were obtained. The magnetic induction increased and the hysteresis loss of the finished sheets decreased when 0.05%P and 0.10%P were added. The 0.05%P added in the steel was recommended to decrease the anomalous loss, but phosphorus added in the steel results in a more complicated domain structure and more hysteresis loss generated in return. Yoshida et al. [13,14] disclosed that phosphorus added to steel reduced the grain boundary energy, and then, the grain growth rate decreased. The addition of phosphorus can promote the distribution of Mn among the dendrites and thus affect the segregation of Mn at the grain boundaries. The precipitation temperature and quantity of MnS decreased during the phase transformation. In conclusion, phosphorus is beneficial to the electromagnetic and mechanical properties, but the appropriate content of phosphorus in the steel should be adjusted based on the composition of the steel grades.

Currently, steel manufacturers have paid more attention to increasing the utilization proportion of scrap resources in the basic oxygen furnace (BOF). Phosphorus will be enriched to a certain extent after adding more scraps into the converter due to the further dephosphorization limitations under the present smelting conditions. Both properties of enhancing and low-cost preponderance can be obtained by using appropriate P-bearing liquid steel to manufacture low-carbon low-silicon NOES. The magnetic properties and stamping performance of P-bearing steels will be improved after the trade-off. Although many researchers [9,11,12] suggested that only a moderate content of phosphorus addition in the steel had the effect of improving magnetic properties and mechanical properties of NOES, they did not give the theoretical foundation on how to design the appropriate content of phosphorus added in the low-grade specified steel. The influence mechanism of phosphorus content on the number density and type of the fine precipitates, microstructure, texture, magnetic properties, and mechanical properties in low-carbon and low-silicon NOES is seldom reported. It is significant to improve the quality of the above-mentioned steel by adding appropriate phosphorus content. Therefore, it is essential to elucidate the effect of regularity and the mechanism of phosphorus on the magnetic properties and mechanical properties of the mentioned steels.

In this work, four targeted phosphorus content levels were designed to add into a specified Al-killed low-carbon low-silicon NOES. The influence of phosphorus content on the precipitation behaviors of the precipitates, microstructure, texture, and magnetic and mechanical properties was investigated in detail. The characteristics of the precipitates, microstructure, and texture in the different P-bearing sheets were compared. The influence mechanism of phosphorus content on the precipitates, microstructure, texture, and magnetic and mechanical properties was elucidated. It is expected to provide a theoretical foundation and reference data for phosphorus content selection and precipitates’ controlling effectively in the industrial production of low-carbon and low-silicon NOES by using P-bearing hot metal.

## 2. Materials and Methods

Referencing the composition control standard of the commercial same grade low-C low-Si NOES, the aimed alloys and impurities of the experimental steels were designed and controlled as Fe-(≤0.0050%)C -(0.50~1.00%)(Si+Al)-(≤0.50%)Mn-(≤0.0020%)S (sulfur)-(≤0.0030%)O (oxygen)-(≤0.0030%)N (nitrogen). Based on the industrial manufacturing route and the segregation characteristics of phosphorus in steel reported in the literature, the target phosphorus content levels in the steel were designed to be 0.06%, 0.10%, 0.14%, and 0.18%. Samples with a target content of P at the 0.06% level are abbreviated as 0.06%P -targeted samples, and the rest are abbreviated in the same manner. The prepared specimens were labeled as 0.06%P, 0.10%P, 0.14%P, and 0.18%P, respectively.

The experimental steel was melted in a vacuum induction furnace, and a 50kg sample was obtained. The melting temperature was 1873 K. The chemical composition of the steel was analyzed and met the above design requirement. The qualified molten steel was cast into ingots in a water-cooled copper mold surrounded by silica sand. The sample was reheated by 1373 K × 90 min and then hot rolling was performed, with a starting hot rolling temperature of 1273 K, a final rolling temperature of 1133 K, to obtain a thickness of 2.5 mm. After, the hot-rolled bands were cold rolled to obtain a sample with a thickness of 0.5 mm, and then, the cold-rolled samples were cut to 30 mm × 300 mm (parallel to the rolling direction) and annealed continuously at 1093 K for 5 min to obtain a 4-component sample with a size of 30 mm × 180 mm × 0.5 mm.

The metallographic analysis pieces with a size of 10 × 7 mm (RD × TD) of the finished samples were prepared after polishing. With the aid of the field emission scanning electron microscope (FE-SEM, ThermoFisher/ApreoSHiVa) combined with an energy dispersive X-ray spectroscope (EDS, AztecUltimLive100X), 100 consecutive fields of view were taken of the experimental steel at 50,000× to observe precipitates of less than 500 nm in the steel. INCA and Image Pro Plus were used to analyze the composition, quantity, and size distribution information of the precipitates in the steel. The quantity density was obtained from Equation (1) and was used to evaluate the number of precipitates. Then, the samples were corroded; the corrosive solution consisted of 4% nitric acid and 96% alcohol (volume fraction); the microstructure was observed under an optical microscope (OM, Axioplan-2 Zeiss), and 5 consecutive field images were taken with a 100 × magnification to calculate the mean grain size with Image Pro Plus. The texture of the samples (16 mm × 15 mm × 0.35 mm) was measured by an X-ray diffractometer (XRD, Bruker D8 Advance). The radiation source was the Cu target; the radiation mode was a point focal spot; the voltage/current in the tube was 40 kV/40 mA. According to the *2θ* values (44.670°, 65.023°, 82.335°) under the diffraction peaks {110}, {200}, and {211} of the ferrite, the incomplete pole figure of the samples was measured in the standard mode with a step length of 5°, where the rotation angle α was 0~75° and β was 0~360°. The TexTools software was used to calculate the orientation distribution function (ODF).
(1)$$N_{A}=\frac{N_{T}}{n\times A_{\mathrm{obs}}}$$
where *N_A_* represents the number density of the precipitates in the unit observation area (#/mm^2^), *N**_T_* represents the total number of precipitates (#), *A*_obs_ represents the single scanning area of the precipitates (μm^2^), and *n* represents the number of field images.

A soft magnetic measuring instrument (MPG100-D) was used to measure the magnetic properties. The magnetic properties of 16 pieces (30 × 180 × 0.50 mm) were measured, and the average value was taken as the result. The magnetic induction (*B*_50_, T) was measured under a magnetic field of 5000 A/m, and the core loss (*P*_15/50_, W∙kg^−1^) was measured when the designed magnetic induction of the sample was 1.5 T under 50 Hz. The four-component steel samples were made into standard tensile samples with a standard distance of 50 mm, and room-temperature tensile tests were performed using an Instron 3382 electronic universal testing machine at a tensile rate of 3 mm/min. The lower yield strength *R_el_*, tensile strength *R_m_*, and elongation after break *A*_50_ of the samples were measured.

## 3. Results and Discussion

### 3.1. Effect of Phosphorus Content on Precipitation Behavior of Precipitates

Based on the analysis results of FE-SEM/EDS, the precipitates in the steels were mainly aluminum (Al)-manganese (Mn)-Si-bearing complex nitrides ((Al,Si,Mn)*_x_*N*_y_*) and P-bearing complex nitrides ((Al,Si,Mn)*_x_*N*_y_*-P). Increasing the target phosphorus content in the steels, (Al,Si,Mn)*_x_*N*_y_* decreased and (Al,Si,Mn)*_x_*N*_y_*-P increased, which resulted in the total precipitates increasing. A minor quantity of Al_2_O_3_ was detected in the 0.06%P-targeted samples. (Al,Si,Mn)*_x_*N*_y_*-MnS-P and (Al,Si,Mn)*_x_*N*_y_*-MnS composite precipitates were detected in the 0.12%P-targeted samples. No single MnS or AlN was detected, and the main precipitates were complex nitrides of Si, Mn, and Al. Typical precipitates smaller than 500 nm detected in the samples are shown in Table 1. The quantitative percentage *N*_F_(%) of the precipitates is defined as the ratio of the detected particle number to the total number of different types of precipitates that occur in the whole statistics area. Hexagonal crystal system structure composite nitrides of Mn, Al, and Si with a Si/Mn atomic ratio of 4 were detected in grain-oriented silicon steel 0.25%Mn-bearing [15]. Al can enter the composite oxynitride of Si and Mn to produce (AlN)*_x_*(SiMn_0.25_N*_y_*O*_z_*)_1*-x*_ (*x* changes from 0 to 1) during the heat treatment process of the steel. In the present work, Al and P content in the steel was much higher than that in grain-oriented silicon steel; (Al,Si,Mn)*_x_*N*_y_*, (Al,Si,Mn)*_x_*N*_y_*-P were easily detected in the steel; the (Al,Si,Mn)*_x_*N*_y_*-MnS-P composite precipitates were detected after increasing the target phosphorus content in the steel, which indicates that the addition of phosphorus in the steel changes the precipitation amount and mode of MnS and AlN. 

The composition of the precipitates of less than 500 nm in the sample is shown in Figure 1. N is a light element; thus, when the N content in the precipitate is low, it does not appear in the energy spectrum diagram. Although Al and Mn in steel can improve resistivity and reduce eddy current loss, the presence of N and S in steel will produce the AlN and MnS fine precipitation phase, which will hinder the grain boundary movement and grain growth during steel recrystallization, increase iron loss, and deteriorate the magnetic properties.

Figure 2 indicates the correlation of the number density and average size of precipitates in the samples with the increase of the target phosphorus content. The number density and the average size of the precipitates increased initially as the target phosphorus content in the steel increased, and then, the lowest number density and the largest average size of the precipitates appeared in the 0.14%P-targeted sample. The average size of the main precipitates in the samples is shown in Figure 3. Figure 4 is the percentage of the number density of precipitates of different sizes of the total precipitates with different P-targeted samples. Combined with the results listed in Table 1, when the target phosphorus content increased, the number density of (Al,Si,Mn)*_x_*N*_y_* in the steel decreased, and the number density of the P-containing precipitates (Al,Si,Mn)*_x_*N*_y_*-P and (Al,Si,Mn)*_x_*N*_y_*-MnS-P tended to increase. The average size of the precipitates decreased when the target phosphorus content increased to 0.18%. This was related to the removal or coarsening of the precipitates after phosphorus increased in the steel. Compared with the 0.14%P-targeted sample, only a small number density increase of the precipitates appeared in the 0.18%P-targeted sample, which indicates more P addition exceeding 0.14% has little effect on the precipitation of fine P-bearing precipitates within the statistical error. However, adding another point beyond 0.14%P would make the results more reliable, and the experimental results were more detailed to certify the effect of P in the steel.

Based on the composition of the experimental samples and the solubility products of MnS, AlN, and Si_3_N_4_ in the ferritic steel in Table 2, the theoretical solid solution or precipitation temperature of MnS, AlN, and Si_3_N_4_ in solid steel are 1493 K, 1438 K, and 1105 K, respectively. There were some precipitate solution, precipitation, and growth during reheating, rolling, and the cold-rolled plate continuous annealing process. In particular, unstable Si_3_N_4_ in fine-sized (Al,Si)*_x_*N*_y_* or (Al,Si,Mn)*_x_*N*_y_* will solubilize or precipitate at 1105 K. The number density and size of the precipitates in the steel changed after the above hot working treatment. In this study, when the target phosphorus content in the steel was different, the precipitation of MnS and AlN was hindered to a different degree by the segregation of phosphorus at the dendritic spacing and grain boundaries during the solidification and hot working treatment of the steel. Thus, the number density of (Al,Si)*_x_*N*_y_* or (Al,Si,Mn)*_x_*N*_y_* precipitates detected in the finished steel decreased with the increasing target phosphorus content. However, phosphorus segregated at the grain boundary easily became the precipitation core of MnS and AlN, and complex phosphides such as (Al,Si,Mn)*_x_*N*_y_*-P and (Al,Si,Mn)*_x_*N*_y_*-MnS-P were generated and remained in the steel matrix. Additionally, more P-containing precipitates with smaller sizes were generated when the target phosphorus content increased, resulting in a decrease in the average precipitate size when 0.18%P-targeted was controlled in the steel.

The precipitates with a size equivalent to the thickness of the magnetic domain wall in NOES have a negative effect on the magnetic properties [1]. The largest pinning effect on the movement of the domain wall during grain growth and magnetization often will appear when there are many fine precipitates (about 100~200 nm [1]) in the steel. When there are large quantities of precipitates of the above size generated in the steel, the hysteresis loss increases, and the magnetic properties of the steel will deteriorate [6]. The research results show that fine precipitates smaller than 500 nm coarsen and their number density decreases after the target P content increases appropriately. It is desired to control the quantity and size of the precipitates by adjusting the phosphorus content in the steel when the content of Al, Mn, S, and N in the steel is stable. From the point of view of controlling fine precipitates in the steel, the reasonable target phosphorus content controlled as 0.14% in the steel can achieve the objective of reducing the number of fine precipitates in the steel and increase their size.

### 3.2. Effect of Phosphorus Content on the Microstructure of the Finished Steel

Figure 5 is the optical micrograph of the finished samples. According to Figure 5, the recrystallization already finished after continuous annealing at 1093 K. The average grain size was small in the P-bearing NOES. The grain size of the samples with the target content of P at the 0.14% level was obviously larger than that of the other phosphorus content samples, and these samples all had a ferrite grain microstructure. The microstructure of the samples with 0.06% and 0.18% target phosphorus content was more homogenous, whereas the samples with 0.10% and 0.14% target phosphorus content had some fine grains among the relatively larger ones. Loss in NOES mainly includes hysteresis loss and eddy current loss. Grain boundaries can prevent magnetic domain walls from moving, and the size and energy loss during moving of the magnetic domains are determined by the grain size [19]. The magnetic properties of NOES are greatly influenced by the distribution of the microstructure, inclusions, and texture composition [1].

The statistical results of the average grain size in different target P-bearing samples of the finished samples are shown in Figure 6. Figure 7 shows the percentage of grain quantity with different sizes of the total grains in different P-targeted samples. According to the results, the average grain size of the samples was 23 µm, 24 μm, 26 μm, and 21 μm with the 0.06%, 0.10%, 0.14%, and 0.18% target phosphorus content in the samples, respectively. The average grain size increased at first and then decreased after the target phosphorus content was attained at 0.18%. Phosphorus segregation at grain boundaries can prevent grain boundaries from migration, which probably prevents grain growth. The standard deviation of the grain size was 13 μm for the 0.06%P-targeted and 0.18%P-targeted samples, while it was 15 μm for the 0.10%P-targeted and 0.14%P-targeted samples. Combined with Figure 7, it is indicated that the microstructure homogeneity of the samples deteriorated slightly as the target phosphorus content increased. It is certain phosphorus inhibited the grain growth and affected the homogeneity of the microstructure of the finished samples, which caused an increase in the grain boundary area, hindered domain movement, and resulted in higher hysteresis loss. In addition, the segregation of phosphorus elements near the dislocations can strongly hinder the movement of the dislocations, which leads to delayed recrystallization of the samples [20].

It was found that the number density of the precipitates decreased and their average size increased after the target phosphorus content in the sample increased to 0.14%. As a result, the pinning force on the grain boundary weakened, which is conducive to grain growth, and the average grain size of the steel increased. A similar result was reported by Li [21]: the precipitation of 0.3~0.7 μm-scale particles at the grain boundary hindered the growth of the grains. On the other hand, the fine precipitates in the samples reduced as a result of phosphorus segregation at the grain boundaries, and part precipitates coarsened, which weakened the pinning force at the grain boundaries or domain wall and increased the grain size, resulting in an improvement in the magnetic properties. To evaluate the effect of precipitates’ characteristics on the grain growth, the pinning force that prevents the grain from growth was approximately estimated by applying the pinning theory proposed by Zener and calculating Formulas (2)–(6) from [22,23,24,25]. The equations are as follows.
(2)$$P_{Z}=\frac{3}{2}\cdot \frac{f_{V}\sigma }{r}$$
(3)$$N_{V}=\frac{2}{\pi }\cdot \frac{N_{A}}{d_{A,\;m}}$$
(4)$$d_{A,\;m}=\frac{N_{T}}{\sum d_{A,\;i}}$$
(5)$$d_{V,\;m}=\frac{\pi }{2}d_{A,\;m}\;$$
(6)$$f_{V}=\frac{\pi }{6}\cdot d_{V,\;m}^{3}\cdot N_{V}$$
where: *P*_Z_ is defined as the Zener pinning force of the particle on the grain boundary per unit grain boundary area. *f*_v_ is the volume fraction of the particle. *σ* is the free energy per unit grain boundary. *r* is the average radius of the second-phase particles. *N*_v_ is the number of particles per unit volume of the grain boundary. *N*_A_ is the number of particles on the grain boundary per unit area. *d*_A, m_ is the mean diameter of the particles on the grain boundary per unit area. *d*_V, m_ is the mean diameter of the particles in the grain boundary per unit volume. *d*_A, i_ is the diameter of the *i*-th particle on the grain boundary per unit area. The more the number of precipitates and the smaller the size, the stronger the inhibition force of grain growth. Assuming only the size and number of precipitates are taken into consideration, the pinning force of the precipitates on the grain boundaries in the four samples can be theoretically estimated based on the statistical results. The increasing order of the values was *P*_Z, 0.18%P_, *P*_Z, 0.06%P_, *P*_Z, 0.10%P_, *P*_Z, 0.14%P_ according to the calculated results *P*_Z, 0.14%P_ = 0.48*P*_Z, 0.06%P_ = 0.21*P*_Z, 0.10%P_ = 0.96*P*_Z, 0.18%P_. The grain boundary pinning force is determined by the quantity, size, and distribution of the precipitates in the samples simultaneously. Although the pinning force of 0.10%P-targeted samples was larger than the other samples, the size of the precipitates in 0.18%P-targeted was more dispersed, as seen in Figure 4, and the dispersed precipitates were more homogeneous. The least pinning force on the grain boundaries generated from the precipitates was found in the 0.14%P-targeted samples, and the grain size was the largest, while the number density was the lowest within the four samples. It is well known that grain growth is controlled by three factors, the driving force stored in the matrix, the resistance of the grain growth, and the pinning force of the inhibitors. The grain size and microstructure homogeneity are influenced simultaneously by phosphorus segregation at the grain boundaries and precipitates’ characteristics in the finished samples. Phosphorus segregation and precipitates’ precipitation at the grain boundary retard grain boundary migration, inhibit grain growth, and deteriorate the microstructure homogeneity of the steel. The target phosphorus content in the steel must be controlled accurately based on the specific composition of NOES to function favorably.

### 3.3. Effect of Phosphorus Content on the Texture of the Finished Steel

In NOES, γ-fiber texture ({111} parallel to the rolling surface has a higher average magnetocrystalline anisotropy. The main texture exhibits an easy magnetization <100> direction on the η-fiber texture (<100> parallel to the rolling direction), and the magnetic induction intensity was obviously affected [26,27].

Figure 8a–d represent ODF sections taken at *φ*_2_ = 45° representing the texture in the samples with different target phosphorus content, respectively. The main texture of the sample is the γ ({111} *‖*RD) texture (*Φ* = 55°, *φ*_2_ = 45°), and the γ texture is distributed continuously. Each sample contains a small amount of {100} *‖*RD(*Φ* = 0°, *φ*_2_ = 45°), {411} *‖*RD (*Φ* = 19°, *φ*_2_ = 45°), {112} *‖*RD texture (*Φ* = 35°, *φ*_2_ = 45°), and {110} *‖*RD texture (*Φ* = 90°, *φ*_2_ = 45°).

Table 3 lists the macroscopic texture content *T*_V_ (%) of the main texture components in different target phosphorus contents of the experimental samples calculated by the texture analysis software. As the target phosphorus content increased, the {111}<112> component decreased. Since the {111}<112> component severely weakened the magnetic induction in electrical steels, phosphorus addition in the steel was favorable. However, the {111}<110> component, which is unfavorable for the magnetic properties of NOES, increased when the target phosphorus content increased to 0.18% compared with that 0.14%-targeted sample. The {001}<110> component (cube texture) also tended to increase with the increase of the target phosphorus content in the samples, which was beneficial to the magnetic properties. The findings indicate that phosphorus was used as a segregated element in NOES, and the texture development of the steel was significantly affected. It is perhaps that {111} recrystallization nucleation close to the grain boundaries is inhibited by phosphorus segregation, which leads to the components of {111}<112> changing, as reported by Lee et al. [9]. The change in the main texture content of the samples indicates the phosphorus element is advantageous for enhancing the magnetic properties of NOES. The increase of the {111}<110> component probably originated from the increase in the grain boundaries caused by more numerous and smaller-sized precipitates, which inhibited the grain growth. The content ratio of the {100} texture to the {111} texture was 1.365 in the samples with the target content of P at the 0.14% level and larger than that in the 0.18%P-targeted sample. As a result, the content of the favorable texture component was the most in the 0.14%P-targeted samples, and then, the magnetic induction of the samples was better.

### 3.4. Effect of Phosphorus Content on the Magnetic Properties of the Finished Steel

Figure 9 shows the effect of the target phosphorus content on the iron loss and magnetic induction of the experimental steels. In the four samples of 0.06%P, 0.10%P, 0.14%P, and 0.18%P, the magnetic induction B50 was 1.742 T, 1.752 T, 1.765 T, and 1.748 T and the iron loss P1.5/50 was 4.53 W/kg, 4.69 W/kg, 4.85 W/kg, and 4.93 W/kg, respectively. The magnetic induction of the samples with a target content of P at the 0.14% level was higher than that of the other samples. Compared with the 0.06%P-targeted samples, the magnetic induction increased by 0.023 T compared to that of the 0.14%P-targeted samples. The decrease in the {111}<112> component and the number density of the precipitates and the increase in their average size were responsible for the increase in the magnetic induction. A decrease in the magnetic induction of the 0.18%P-targeted sample compared with the 0.14%P-targeted sample was caused by grain refining and an increase of the unfavorable {111}<110> component. Iron loss increased as the target phosphorus content increased in the steel, including eddy current loss and hysteresis loss. The grain size increased and the homogeneity of the microstructure deteriorated when the target P content increased from 0.06% to 0.14%. Thus, higher core loss appeared in the samples with larger average grain sizes in the present study. It is the reciprocal relationship between hysteresis loss and the square root of the grain size that increases the hysteresis loss [28,29,30]. The changes of the magnetic induction are proportional to the coercive force, while that of the core loss is proportional to the hysteresis loss [1]. The coercive force is directly proportional to the number of precipitates and inversely proportional to the size of the precipitates. When the size of the precipitates was near the domain wall thickness (100~200 nm), their impact on the coercive force and hysteresis loss was the biggest, and the ability to pin the domain wall was the strongest. After P was added to the steel, the number density of the precipitates (in the size range of 100~200 nm) increased and then reduced to the least in the 0.14%P-target samples (Figure 2), which can decrease the coercive force and hysteresis loss of the specimens. On the other hand, phosphorus added to the steel can increase resistivity and reduce eddy current losses. Due to the tendency of the fine precipitates in the steel pinning the grain boundaries, recrystallization was delayed and grain growth was inhibited, which increased the hysteresis loss and degraded the magnetic properties.

### 3.5. Effect of Phosphorus Content on Mechanical Properties of the Finished Steel

Figure 10 shows the effect of the target phosphorus content on the mechanical properties of the experimental steel. In the four samples, that target phosphorus content was 0.06%, 0.10%, 0.14%, and 0.18%, the tensile strength was 385 MPa, 392 MPa, 408 MPa, and 423 MPa, the elongation was 40%, 39%, 38%, and 35%, and the lower yield strength was 248 MPa, 252 MPa, 270 MPa, and 283 MPa, respectively. The tensile and lower yield strengths increased and the elongation decreased with the increase of the target phosphorus content in the samples. The tensile strength and the lower yield strength were improved by 38 MPa and 35 MPa, and the elongation was decreased by 5% when the target phosphorus content in the steel increased from 0.06% to 0.18%. The atomic radius of phosphorus is smaller than the matrix element Fe and the alloy element Si in NOES, which results in solid solution strengthening in the finished steel. The degree of solid solution strengthening increased with the target phosphorus content in the samples. On the other hand, when the target phosphorus content increased, the grain growth was restricted and the grain size was refined, which increased the strength at the expense of some of its elongation. In the present work, solid solution strengthening seemed to have the main effect on the strength of the finished steel compared with the influence of the grain size caused by the precipitates.

## 4. Conclusions

(1) The number density and average size (100~200 nm) of the precipitates in the steel were influenced by the addition of phosphorus. The main precipitates were (Al,Si,Mn)*_x_*N*_y_* and (Al,Si,Mn)*_x_*N*_y_*-P in the steels with different target phosphorus contents, and minor quantities of (Al,Si,Mn)*_x_*N*_y_*-MnS-P, (Al,Si,Mn)*_x_*N*_y_*-MnS and Al_2_O_3_ were also detected. The number density of the precipitates was the lowest and the average size of the precipitates was the largest when the target phosphorus content in the steel was 0.14%.

(2) The average grain size and microstructure homogeneity of the steel were influenced by the addition of phosphorus. The average grain size was small in the P-bearing NOES. The phosphorus segregation and precipitates’ precipitation at the grain boundary retarded grain boundary migration, inhibited grain growth, and deteriorated the microstructure homogeneity of the steel. The content of the {111}<112> component decreased and the favorable texture increased after phosphorus was added to the steel. The most content ratio of the {100} favorable texture to the unfavorable {111} texture was 1.365, and the biggest grain size was 26 μm in the samples with a target content of P at 0.14% level.

(3) The magnetic induction increased by phosphorus, inhibiting the recrystallization nucleation of the {111} texture near the grain boundary. The grain refinement and deterioration of the microstructure homogeneity also led to a small increase in iron loss with the increase of the target phosphorus content in the samples. The maximum magnetic induction (*B*_50_ = 1.765T) appeared in the samples with a target content of P at a 0.14% level, in which the iron loss *P*_1.5/50_ was 4.85 W/kg. Phosphorus can influence the quantity and average size of the fine precipitates and change the microstructure and texture of the steel, which influence the magnetic properties of the finished steel. The magnetic induction property of the investigated steel was superior to the same commercial-grade products.

(4) The tensile strength was improved by 38 MPa; the lower yield strength was increased by 35 MPa; the elongation was decreased by 5% when the target phosphorus content in the steel increased from 0.06% to 0.18%. The mechanical properties of the steel were mainly improved by solid solution strengthening and the grain refining effect of the phosphorus added to the steel under elongation, meeting the performance requirement. The strength of the investigated steel was superior to the same-grade products.

(5) Moderate P added in low-grade NOES can affect the number density and average size of the precipitates. The effect of the average grain size on the iron loss of the samples presented a reverse trend with the targeted P content increasing from 0.06% to 0.14%, owing to the microstructure difference arising from the P content under the same experimental conditions. Moderate P added in low-grade NOES can improve the magnetic induction and mechanical properties. The recommended target P content in the steel was no higher than 0.14%, ensuring obtaining a homogeneous microstructure and larger grain size in the present NOES.

## Figures and Tables

**Figure 1 materials-15-06332-f001:**
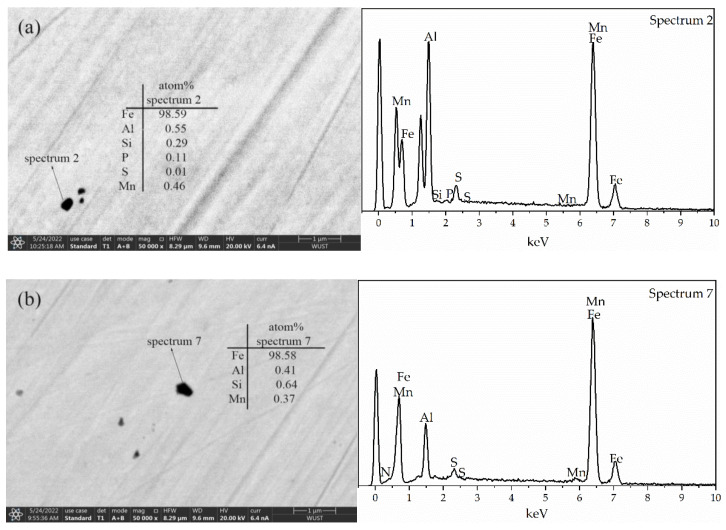
Results of typical precipitates from FE-SEM/EDS analysis in experimental steel: (**a**) (Al,Si,Mn)*_x_*N*_y_*-P; (**b**) (Al,Si,Mn)*_x_*N*_y_*.

**Figure 2 materials-15-06332-f002:**
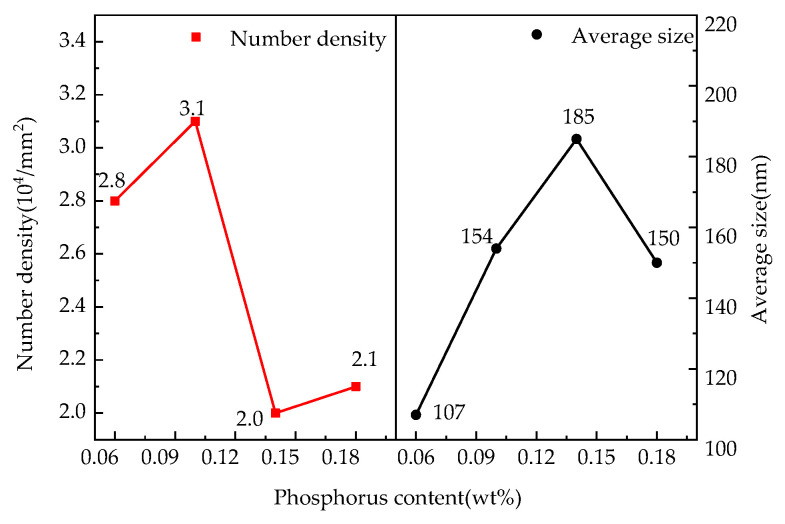
The number density and the average size of precipitates detected in the different P-bearing samples.

**Figure 3 materials-15-06332-f003:**
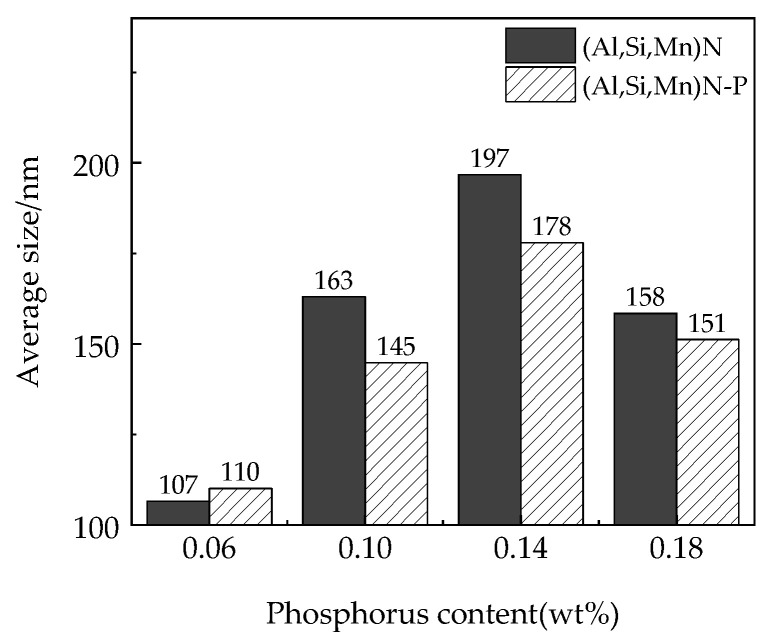
Effect of target P content on the average size of the main precipitates in the finished steel.

**Figure 4 materials-15-06332-f004:**
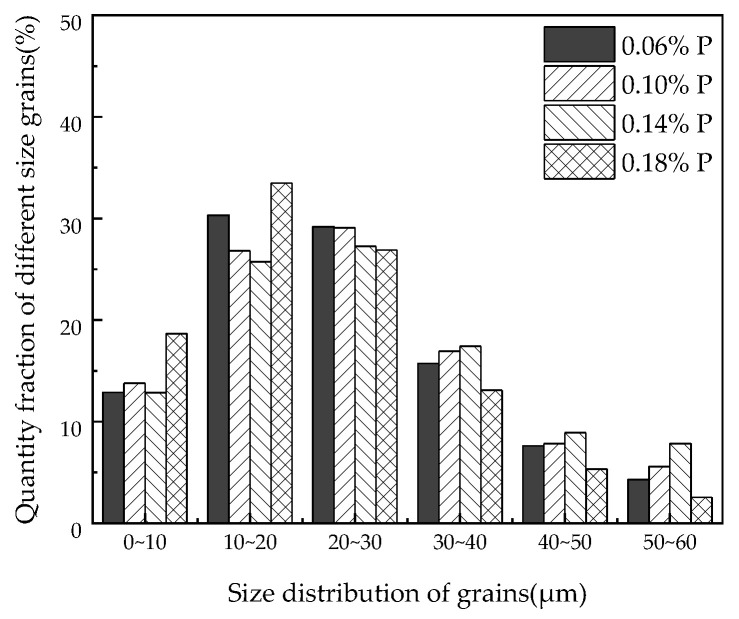
Quantity fraction of different sizes of precipitates in the different P-targeted samples.

**Figure 5 materials-15-06332-f005:**
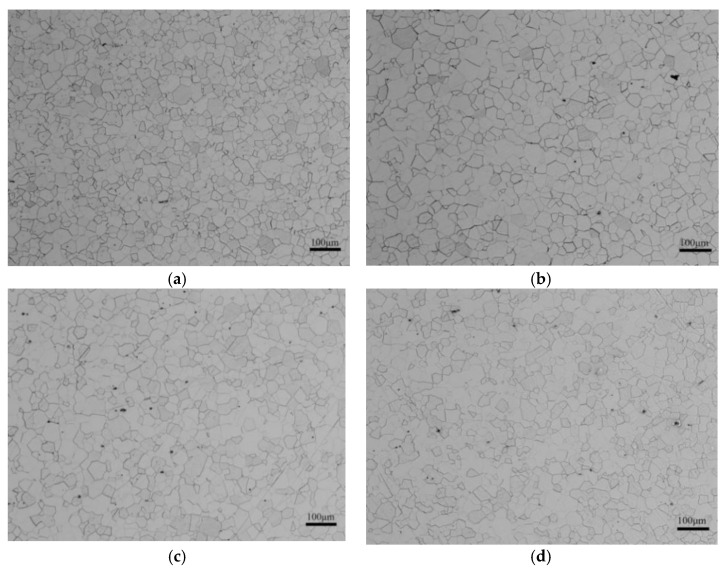
Microstructure of experimental steels: (**a**) 0.06%P, (**b**) 0.10%P, (**c**) 0.14%P, and (**d**) 0.18%P.

**Figure 6 materials-15-06332-f006:**
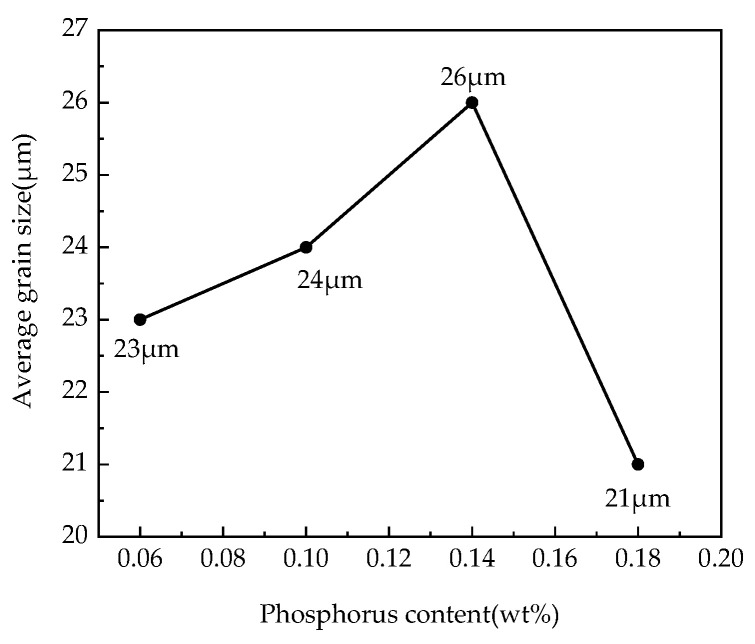
Effect of target P content in the samples on average grain size.

**Figure 7 materials-15-06332-f007:**
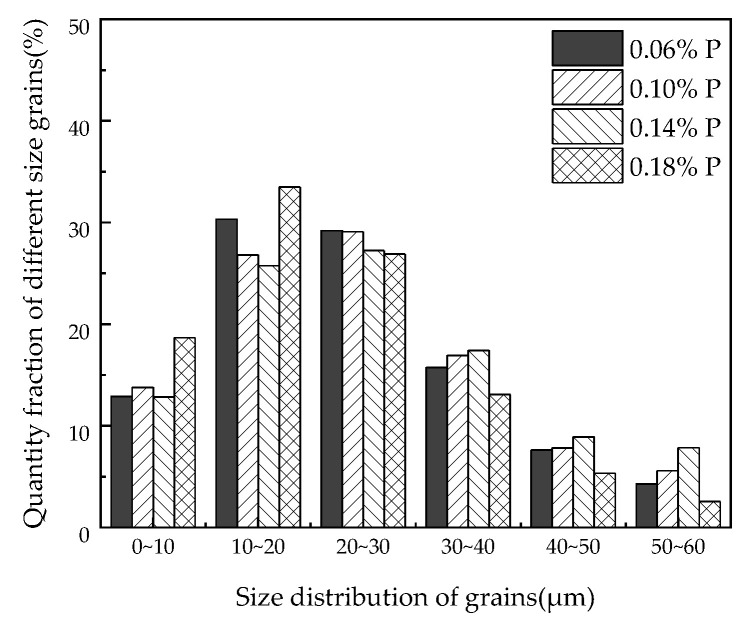
Quantity fraction of different size grains in the different P-targeted samples.

**Figure 8 materials-15-06332-f008:**
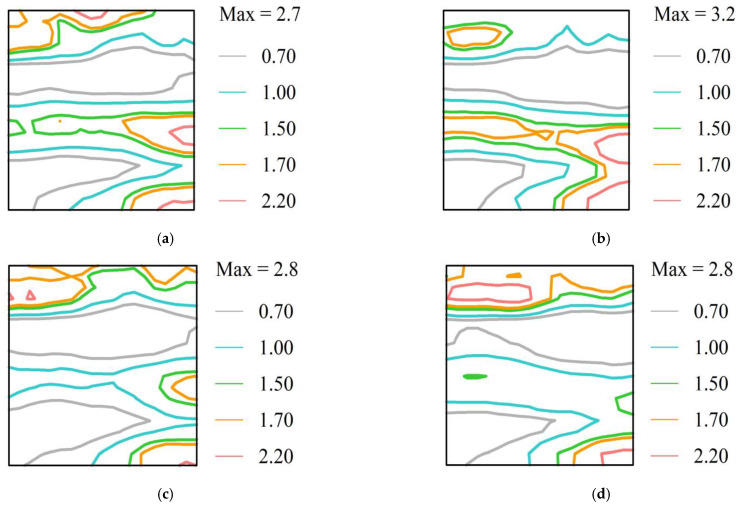
φ_2_ = 45°section of ODF representing the texture for different target P content samples: (**a**) 0.06%P, (**b**) 0.10%P, (**c**) 0.14%P, and (**d**) 0.18%P.

**Figure 9 materials-15-06332-f009:**
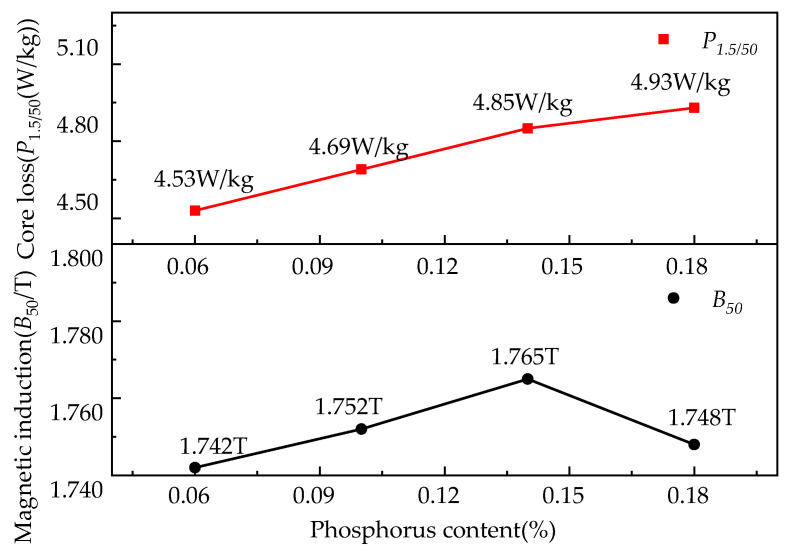
Effect of target P content on iron loss and magnetic induction.

**Figure 10 materials-15-06332-f010:**
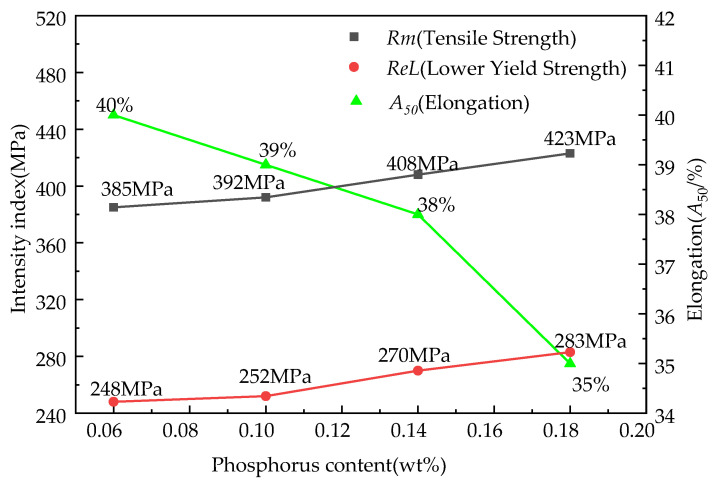
Effect of the target P content on the strength and elongation of the samples.

**Table 1 materials-15-06332-t001:** Typical composition and percent quantity of major precipitates in the continuously annealed samples, *N*_F_(%).

Sample	(Al,Si,Mn)*_x_*N*_y_*	(Al,Si,Mn)*_x_*N*_y_*-P	(Al,Si,Mn)*_x_*N*_y_*-MnS-P	(Al,Si,Mn)*_x_*N*_y_*-MnS	Al_2_O_3_
0.06%P	78.0	18.0	-	-	4.0
0.10%P	46.1	53.9	-	-	-
0.14%P	16.4	65.7	12.3	5.6	-
0.18%P	5.6	93.5	0.9	-	-

**Table 2 materials-15-06332-t002:** Thermodynamics data of MnS, AlN, and Si_3_N_4_ in the ferrite steel.

Equation	Ferrite (α)
${[\mathrm{Mn}]+[S]=\mathrm{MnS}}_{\left(S\right)}$	$\log(w[\mathrm{Mn}]\cdot w[S])=-10590/T_{\alpha }+4.092^{}$ [16]
${[\mathrm{Al}]+[N]=\mathrm{AlN}}_{\left(S\right)}$	$\log(w[\mathrm{Al}]\cdot w[N])=-11420/T_{\alpha }+5.12^{}$ [17]
$4{\left[N\right]}_{\left(\alpha -Fe\right)}+3[\mathrm{Si}]={\mathrm{Si}}_{3}N_{4}{}_{\left(s\right)}$	$\Delta G^{0}=-595860+377.08T_{\alpha }(J/mol)$ [18]

**Table 3 materials-15-06332-t003:** The content of the main texture components in different target P contents of the experimental steel (volume ratio/%, deviation angle < 15°).

Sample	T_V_ (%)					$\frac{\left\{100\right\}}{\left\{111\right\}}$
	{100}	{110}	{111}	{112}	{411}	
0.06%P	0.145	0.247	0.156	0.297	0.357	0.929
0.10%P	0.134	0.245	0.164	0.313	0.346	0.817
0.14%P	0.168	0.242	0.123	0.281	0.397	1.365
0.18%P	0.164	0.246	0.128	0.284	0.383	1.281

## Data Availability

Not applicable.

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
