# Peer review of "Effect of Phosphorus Content on Magnetic and Mechanical Properties of Non-Oriented Electrical Steel"

_materials, 2022, doi:10.3390/ma15186332_

Round 1

Reviewer 1 Report

I fear the number of data points used to prepare the paper is too low for the complex manufacturing process studied in the paper. If they showed a steady trend the conclussions could be really meaningful, but in this case the trend figures show is broken when P content reaches 0.18% (figure 2a evolution is even odder): I'm not saying the results are wrong, but the authors should consider repeating the test for 0.18%P or adding another data point as some results could be due to statistical dispersion.

English must be also careafully edited, as some of the problems found could be due to languaje inaccuracies. For example:

Line 205: Change "...least..." to "...lowest..."

Line 88: Define the meaning of "BOF". Basic Oxygen Furnace?.

lines 94-96: "...many researchers...". Please, give some references.

line 121. Change "...018..." to """...0.18..."

Line 244: A reference is needed

Lines 262-265: I suppose the authors are talking about grain size when they refer to homogeneity, but figure 7 show a very similar distribution of grain sizes for all samples. inhomogeneity should be justified with better data. Or are the authors refering to precipitates distribution?

Lines 285-287: The authors should explain why they consider the microstructure deteriorates as P content increases because figure 7 doesn't show significant differences in grain size distributions nor is the difference between 13 and 15 microns enough

Lines 287-289: The authors state P inhibits grain growth , but figure 6 says otherwise for P contents below 0.18%. Also, if more grain boundaries are straightly associated with higher magnetic losses, the higher the grain size, the lower the magnetic losses, and that is something the graphs don't show

Line 315: The value of Pz for 0.18%P is missing. Also, reorder the values from 0.06% to 0.18%P

Line 315: Add the formula for the pinning force

Line 366: "As a result, the favorable texture is the most and magnetic...". There seems some text is missing between "most" and "and magnetic"

Lines 374-376: "Obviously, the magnetic induction of samples with target content of P at 0.14% level is higher than other samples with 0.14 %P". I suppose the authors want to say "...is higher that the other samples".

Lines 383-384: "As the target phosphorus content increases, the grain size increases while the homogeneity of the microstructure deteriorates", but for 0.18%P grain size diminishes, so, the sentence is not true.

Lines 384-385: "Thus, the average grain size is increases, both the magnetic induction and the core loss increase in the present study". Please, change "increases" to "increased" or, better, rewrite the sentence. Also, for a P content of 0.18% the behaviour of core loss and magnetic induction show an opposed behavior.

Reviewer 2 Report

The authors studied the effect of target phosphorus content on precipitates, microstructure, magnetic and mechanical properties of low carbon and low silicon steel.

 There are several problems to be addressed:

 Figures 4 and 7 are difficult for understanding. The authors should transform it in separate graphs for each P-targeted sample.

 In Fig. 5, the optical micrograph of the samples with higher magnifications should be added to show the difference between surface structures.

 There are several publications related to this investigation. Include few references from “Materials”.

 In Fig. 10 the mechanical characteristics (Rm,ReL, A50) designation should be added  in the figure signature.

 I also suggest to improve the conclusion section. The authors should add some more comments regarding the overall evaluation of the experimental results. Can the authors recommend the use of steel with target content of P at 0.14or additional testing is required?

Reviewer 3 Report

The influence of phosphorus content on the magnetic properties of commercial same grade low C low Si NOES.
Samples were measured for magnetic properties, yield strength Rel, tensile strength Rm, and elongation after break A50.
The research results show that fine precipitates smaller than 500 nm coarsen and their number density decreases after target P content increase appropriately.
The fine precipitates with a size similar to the thickness of the domain wall, about 100~200 nm, have the greatest pinning effect on the moving of the domain wall during the grain growth and magnetization.
The degree of solid-solution strengthening is creased with target phosphorus content in the samples.
It was found that the tensile and lower yield strengths increase, and the elongation decreases with the increasing of target phosphorus content in the samples.
When the target phosphorus content increases, grain growth is restricted and grain size is refined.
The magnetic induction is increased by phosphorus inhibiting the recrystallization nucleation of {111} texture near the grain boundary.
he conducted research made it possible to characterize the structure and properties of the tested steel.
The obtained results constitute a new contribution to the field of materials engineering.
The designed studies were carried out correctly.
The authors presented the conclusions on the basis of the obtained own results.
Drawings are clear and made in accordance with the rules of editing.
The article is clearly written and contains all the relevant information to enable correct conclusions to be drawn.
In my opinion, the article can be published as is.

Reviewer 4 Report

In this research, the effect of target phosphorus content on precipitates, microstructure, texture, magnetic properties, and mechanical properties of low carbon and low silicon non-oriented electrical steel was investigated and the influence mechanism was clarified. The work is good, and it is a new point and I recommend accepting it, but there are some points that must be taken into consideration first to improve the paper.

1- In lines 60 and 360: Edit writing reference from "Lee [8] et al." to " Lee et al. [8]".

2- In lines 58, 68, 73, and 79: Need to add a point to " et al.".

3-A section should be added showing all the data and details for the axial tensile tests to obtain the mechanical properties of the samples and the figures for the relationships between stress and strain.

Round 2

Reviewer 1 Report

The authors have misunderstood my comments regarding the need to add a new data point to the study or to repeat the study of the 0.18%P sample. I was not referring to the repetition of the measurement of the values, but to the preparation of another sample (at the furnace).

This was to remove the possibility of the change in the trend of figures being caused by a problem in the manufacturing of the 0.18%P sample, as a major part of the paper is based only on the last point of the graphs. Adding another point beyond 0.14%P would give the results a much higher reliability.

Nevertheless, if the other reviewers and the editors consider the paper can be published I will only require the authors to add a word of caution about the low number of data points.

Reviewer 4 Report

I recommend accepting it.
